# Comparative Draft Genomes of *Leishmania orientalis* Isolate PCM2 (Formerly Named *Leishmania siamensis*) and *Leishmania martiniquensis* Isolate PCM3 from the Southern Province of Thailand

**DOI:** 10.3390/biology11040515

**Published:** 2022-03-27

**Authors:** Pornchai Anuntasomboon, Suradej Siripattanapipong, Sasimanas Unajak, Kiattawee Choowongkomon, Richard Burchmore, Saovanee Leelayoova, Mathirut Mungthin, Teerasak E-kobon

**Affiliations:** 1Department of Genetics, Faculty of Science, Kasetsart University, Bangkok 10900, Thailand; pornchai.an@ku.th; 2Omics Center for Agriculture, Bioresources, Food and Health, Kasetsart University, Bangkok 10900, Thailand; 3Department of Microbiology, Faculty of Science, Mahidol University, Bangkok 10400, Thailand; suradej.sir@mahidol.ac.th; 4Department of Biochemistry, Faculty of Science, Kasetsart University, Bangkok 10900, Thailand; sasimanas.u@ku.th (S.U.); kiattawee.c@ku.th (K.C.); 5Glasgow Polyomics, College of Medical, Veterinary and Life Sciences, University of Glasgow, Glasgow G12 8QQ, UK; richard.burchmore@glasgow.ac.uk; 6Department of Parasitology, Phramongkutklao College of Medicine, Bangkok 10400, Thailand; l.saovanee@pmk.ac.th (S.L.); mathirut@pcm.ac.th (M.M.)

**Keywords:** *Leishmania orientalis*, *Leishmania siamensis*, *Leishmania martiniquensis*, comparative genomics, bioinformatics, leishmaniasis

## Abstract

**Simple Summary:**

This study successfully sequenced the draft genomes of the southern isolates of *Leishmania orientalis* and *Leishmania martiniquensis* in Thailand. The comparison with the genomes of the northern isolates revealed species-level similarity with a level of genome and proteome variation, suggesting the different emerging strains. Comparing the proteins of these southern strains to those of the northern ones and 14 other *Leishmania* species showed six protein groups with numerous unique proteins: 53 for the southern strain PCM2 of *L. orientalis* and 97 for the strain PCM3 of *L. martiniquensis*. Some of these proteins were related to virulence, drug resistance, drug target, and stress response, which could be targeted for further experimental characterization. Therefore, the findings could initiate further genetic and population genomic investigation, and the close monitoring of *L. orientalis* and *L. martiniquensis* in Thailand and neighboring regions.

**Abstract:**

(1) Background: Autochthonous leishmaniasis, a sandfly-borne disease caused by the protozoan parasites *Leishmania orientalis* (formerly named *Leishmania siamensis*) and *Leishmania martiniquensis*, has been reported for immunocompromised and immunocompetent patients in the southern province of Thailand. Apart from the recent genomes of the northern isolates, limited information is known on the emergence and genetics of these parasites. (2) Methods: This study sequenced and compared the genomes of *L. orientalis* isolate PCM2 and *L. martiniquensis* isolate PCM3 with those of the northern isolates and other 14 *Leishmania* species using short-read whole-genome sequencing methods and comparative bioinformatic analyses. (3) Results: The genomes of the southern isolates of *L. orientalis* and *L. martiniquensis* were 30.01 Mbp and 32.39 Mbp, and the comparison with the genomes of the northern isolates revealed species-level similarity with a level of genome and proteome variation, suggesting the different strains. Comparative proteome analysis showed six protein groups with 53 unique proteins for the strain PCM2 and 97 for the strain PCM3. Certain proteins were related to virulence, drug resistance, and stress response. (4) Conclusion: Therefore, the findings could indicate the need for more genetic and population genomic investigation, and the close monitoring of *L. orientalis* and *L. martiniquensis* in Thailand and neighboring regions.

## 1. Introduction

Leishmaniasis is caused by an intracellular kinetoplastid protozoan, *Leishmania*, in tropical and subtropical areas. The disease is transmitted by the bite of a phlebotomine female sandfly. The promastigotes live in the digestive tracts of the sandflies, while the intracellular amastigotes are within the macrophages and other phagocytic cells of their vertebrate hosts [1]. Four subgenera of this parasite have been described, *Leishmania* (*Leishmania*), *Leishmania* (*Viannia*), *Leishmania* (*Mundinia*), and *Leishmania* (*Sauroleishmania*) [2,3]. WHO estimates that 1.3 million cases are affected by the parasites, and 20,000–40,000 deaths occur worldwide [4]. Several risk factors enhance the spread of leishmaniasis, including poverty, malnutrition, population mobility, poor hygiene, immunocompromised states, and environmental and climate changes [5]. Manifestations can be grouped into cutaneous/tegumentary leishmaniasis (CL) and visceral leishmaniasis (VL) forms [3,5]. CL can be further classified into several manifestations: localized cutaneous leishmaniasis (LCL), mucosal leishmaniasis (ML), disseminated leishmaniasis (DL), and diffuse cutaneous leishmaniasis (DCL). CL is an infection of the skin macrophages, causing localized symptoms to a moderate number of disseminated cutaneous lesions, or more widespread diffuse cutaneous diseases. VL or kala-azar is the severe form, wherein the parasites are released into the blood circulation and multiply within the mononuclear phagocytes of the reticuloendothelial system of the liver, spleen, bone marrow, lymph nodes, and intestine. Host immunity and the species of the parasites cause variations in the VL symptoms, ranging from asymptomatic to death [6]. ML occurs when the parasites spread to the nose, mouth, and throat mucous membranes, causing severe mucosal destruction [3].

More than twenty *Leishmania* species are pathogenic to humans, and 10 of these are of major public health importance [2]. Leishmaniasis is not endemic but has been established as an emerging disease in Southeast Asia [7,8]. Sporadic imported cases have been reported in Thailand for less than 50 years [9,10]. Autochthonous VL cases in Thailand have continuously been reported in the southern province since 1996 [11] and the northern province since 2012 [12]. Leelayoova et al. and Bualert et al. reported that two *Leishmania* samples isolated from two autochthonous VL patients in the south of Thailand were a new species of the *Leishmania enriettii* species complex, namely *Leishmania siamensis* (Trang lineage (TR), isolate PCM2), later renamed to *Leishmania orientalis* in 2019, and another previously reported species in the same subgenus, *Leishmania martiniquensis* (Phang Nga lineage (PG), isolate PCM3), by using 15 isoenzyme analyses and phylogenetics of *hsp70* and *cytB* genes [9,13,14,15]. Between 2015 and 2016, Mungthin and colleagues found co-infection evidence of *L. orientalis* and *L. martiniquensis* with *Leishmania donovani*, *Leishmania lainsoni*, and *Leishmania major* in some asymptomatic AIDS patients, raising concerns regarding these new species in Thailand [16]. As *L. orientalis* and *L. martiniquensis* are closely related species, several previously identified *L. orientalis* or *L. siamensis* isolates in 15 published articles were rechecked and later reclassified as *L. martiniquensis* by microscopic observation, direct agglutination test, indirect immunofluorescence assay, and nucleotide sequence analysis of the ITS1 region, SSU-rRNA, and *hsp70* genes [9]. However, the genetic variability of these two *Leishmania* species remains unexplained, suggesting the need for public health awareness on the possible emergence of Leishmanial diseases and the effectiveness of the existing clinical diagnostic tests and anti-Leishmanial drugs.

Progress in next-generation sequencing is increasingly applied to investigate *Leishmania* species associated with different human diseases. The first genome of *L. major* (causing CL) was published in 2005 [17], followed by those of *Leishmania infantum* (causing VL) and *Leishmania braziliensis* (causing LCL, and sometimes ML) [18]. Another 17 *Leishmania* genomes were deposited in the NCBI genome database with different levels of completeness, including *L. donovani*, *L. mexicana*, *L. tropica*, *L. aethiopica*, *L. panamensis*, *L. enriettii*, *L. amazonensis*, *L. arabica*, *L. gerbilli*, *L. turanica*, *L. chagasi*, *L. guyanensis*, *L. peruviana*, *L. torentolae*, *L. adleri*, *L. lainsoni*, and *L. peruviana* [19]. Generally, *Leishmania* genomes range from 29 to 33 Mb in size (approximately 8300 genes) and are organized into 34 to 36 chromosomes, with aneuploidy observed in some species. Recently, the genomes of *L. orientalis* isolate LSCM4 and *L. martiniquensis* isolate LSCM1 from the northern province of Thailand have been sequenced by using several Illumina and Nanopore sequencing platforms [20]. This *Leishmania* genomic information provides an understanding and basis for the parasite genetics and pathogenesis. Conservation of the gene content and gene order has been shown within the genomes of *Leishmania* species [18]. Most of these genes are densely arranged in a long array of polycistronic gene clusters. Approximately 85% of the *Leishmania* genomes are also similar to those of other kinetoplastids [21]. A small number of species-specific genes (around 12%) are restricted to the genus *Leishmania* and have been important targets for understanding host–parasite interactions and anti-leishmania drug development, e.g., PSA-2 or GP46 family protein, which helps binding to the host macrophages and avoids complement-mediated lysis [21]. Some species, including *L. braziliensis,* have transposable DNA elements (retroposons and telomeric-associated transposable elements, TATEs) and RNAi machinery, which could modify existing genes and regulatory elements, whereas the others, such as *L. major* and *L. infantum*, do not have any [21]. Moreover, different copy number variations and repeated DNA sequences (microsatellites, long terminal repeats (LTRs), long interspersed nuclear elements (LINEs), and short interspersed nuclear elements (SINEs)) are also observed in the genomes of *Leishmania* species [20], possibly leading to resistant phenotypes, as well as those with single-nucleotide polymorphisms (SNPs) and small indels [22]. Comparative genome analysis of *L. peruviana* and *L. braziliensis* revealed a number of differences in chromosome copy number variations (disomic in *L. peruviana* and trisomic in *L. braziliensis*), patterns of SNPs, and indels, suggesting their contribution to the low-pathology phenotype of *L. peruviana* [23]. As the northern and southern Thai isolates of *L. orientalis* and *L. martiniquensis* are several hundred kilometers apart, comparative genomics will provide a better understanding of these two *Leishmania* species’ diversity and their pathogeneses. The genes restricted to these isolates and the two species will allow us to further develop new anti-leishmania drugs and improve the efficiency of the diagnostic tests.

## 2. Materials and Methods

*L. orientalis* isolate PCM2 and *L. martiniquensis* isolate PCM3 samples collected from patients in the southern province of Thailand since 2010 were kindly maintained and provided by the Department of Parasitology, Phramongkutklao College of Medicine, Bangkok, Thailand [13]. Species identification of the isolates PCM2 and PCM3 was previously characterized by enzyme profiles, zymodeme, and molecular typing [9]. Genomic DNA was extracted from these two samples using a genomic DNA extraction kit (Qiagen, Hilden, Germany). DNA quality and quantity were checked by measuring absorbance at 260 and 280 nm using the NanoDrop One/One^C^ Microvolume UV-Vis Spectrophotometer (Thermo Scientific, Waltham, MA, USA), and agarose gel electrophoresis before visualization under a UV transilluminator (Biorad Gel Doc XR Imaging System, Hercules, CA, USA).

Genomic libraries of 150–350 bp were prepared and sequenced by the Illumina GAIIx and HiSeq next-generation sequencing platforms (NovageneAIT, Singapore). Sequence reads were checked for quality control by FastQC program version 0.11.9 (https://www.bioinformatics.babraham.ac.uk/projects/fastqc/), (7 June 2021) [24] before assembling to contigs by de novo assembly using MEGAHIT program version 1.2.9 (https://github.com/voutcn/megahit), (9 June 2021) [25] or reference-assisted assembly using BWA program version 0.7.17 (http://bio-bwa.sourceforge.net/), (21 July 2021) [26]. The genomes of *L. orientalis* isolate LSCM4 (accession number: GCA_017916335.1) and *L. martiniquensis* isolate LSCM1 (accession number: GCA_017916325.1) were used as references. The draft genomes of the southern isolates PCM2 and PCM3 were aligned to examine conserved genomic regions with the reference genomes of the isolates LSCM1 and LSCM4 and visualized using Mauve program version 2.4.0 (http://darlinglab.org/mauve/mauve.html), (5 February 2022) [27].

Genomic variants of the southern isolates of *L. orientalis* (PCM2) and *L. martiniquensis* (PCM3) were called by comparison with the genomes of the northern isolates LSCM1 and LSCM4 using GATK program version 4.2.4.1 (https://github.com/broadinstitute/gatk/releases/tag/4.2.4.1) (5 February 2022) [28], and the VCF files were visualized by the CMplot R package version 3.7.0 (https://github.com/YinLiLin/CMplot), (20 February 2022) [29]. The mapping coverage of southern-isolate sequence reads (PCM2 and PCM3) along the referenced genome of the northern ones (LSCM1 and LSCM4) was calculated with Bowtie2 program version 2.4.2 (http://bowtie-bio.sourceforge.net/bowtie2/index.shtml), (21 July 2021) [30] and visualized by using the Samtools version 1.14 (http://www.htslib.org/), (21 July 2021) [28] to estimate the completeness level of the obtained genomes, which employed lesser genome sequencing platforms compared to those of the northern isolates. The called variants were annotated by SnpEff program version 5.0 (http://pcingola.github.io/SnpEff/), (29 November 2021), to identify single-nucleotide polymorphisms (SNPs), insertions, and deletions (indels) and categorize the variant effects.

Genomic contigs of the southern isolates PCM2 and PCM3 of these two *Leishmania* species were processed through gene prediction using Augustus program version 3.4.0 (https://bioinf.uni-greifswald.de/webaugustus/), (31 March 2021), [31] with the soft-masking parameter for the repeated regions. The coding sequences were functionally annotated by BLASTp (https://blast.ncbi.nlm.nih.gov/Blast.cgi?PAGE=Proteins), (31 March 2021) [32] and PANNZER (http://ekhidna2.biocenter.helsinki.fi/sanspanz/), (21 July 2021) [33] programs for functional assignment and gene ontology (GO) classification. Encoded proteomes of *L. orientalis* isolate PCM2 and *L. martiniquensis* isolate PCM3 were compared with the proteomes of the northern isolates LSCM1 and LSCM4, and those of 14 other *Leishmania* species, including *L. major* (GCA_000002725.2), *L. donovani* (GCA_003719575.1), *L. mexicana* (GCA_000234665.4), *L. infantum* (GCA_000002875.2), *L. aethiopica* (GCA_003992445.1), *L. brazilliensis* (GCA_000002845.2), *L. panamensis* (GCA_000755165.1), *L. chagasi* (GCA_014466975.1), *L. amazonensis* (GCA_005317125.1), *L. guyanensis* (GCA_003664525.1), *L. arabica* (GCA_000410695.2), *L. enriettii* (GCA_017916305.1), *L. tropica* (GCA_014139745.1), and *L. lainsoni* (GCA_003664395.1), which were available from the NCBI genome database, using the bidirectional BLASTp method. The results were formatted and visualized using the R scripts to address the similarities at the proteome level.

## 3. Results

### 3.1. Characteristics of the Draft Genomes of L. orientalis Isolate PCM2 and L. martiniquensis Isolate PCM3

The estimated size of the draft genomes of *L. orientalis* isolate PCM2 and *L. martiniquensis* isolate PCM3 from the southern province of Thailand was approximately 30.01 Mbp, with an overall % GC of 59.02 for *L. orientalis* and 32.39 Mbp with % GC of 59.92 for *L. martiniquensis*, similar to the values reported for other *Leishmania* species (Table 1) [19]. Gene prediction from these two genomes produced 8989 genes for *L. orientalis,* and 9577 genes for *L. martiniquensis*, which resembled those of the northern isolates of *L. orientalis* isolate LSCM4 (8162 genes) and *L. martiniquensis* isolate LSCM1 (7993) [20]. These genomic drafts were submitted to the NCBI genome database with the BioProject accession number PRJNA741905 for *L. orientalis* isolate PCM2 and PRJNA728409 for *L. martiniquensis* isolate PCM3.

Comparative alignment of the genomic contigs from the draft genomes of *L. orientalis* isolate PCM2 and *L. martiniquensis* isolate PCM3 to those of the northern isolates LSCM4 and LSCM1 in Thailand showed similar genomic syntenic blocks and rearrangements (Figure 1). The genomes of these two *L. martiniquensis* had highly similar structures, while those of the two *L. orientalis* isolates were more distinct. The genome of the northern isolate LSCM4 was larger than that of the southern isolate PCM2, indicating certain variable areas to be further investigated. Comparison of 212,887,624 sequence reads of the isolate PCM2 to the genome of the isolate LSCM4 as a reference containing 97 genomic scaffolds and contigs revealed that 205,226,953 reads (96%) could be mapped to 94 contigs of the LSCM4 reference with the coverage percentage above 99% and mean coverage more than 100× (Appendix A). Only three short contigs (JAFHLR010000049.1 (843 bp), JAFHLR010000070.1 (11.3 Kbp), and JAFHLR010000098.1 (1.3 Kbp)) of the LSCM4 reference had a coverage percentage lower than 99% (89%, 95%, and 43%, respectively), confirming the same *L. orientalis* species. Comparison of 152,348,606 sequence reads of the isolate PCM3 to the genome of the isolate LSCM1 as a reference containing 42 genomic scaffolds and contigs revealed that 149,223,676 reads (98%) could be mapped to all scaffolds with a coverage percentage above 99% and mean coverage more than 100× (Appendix A), confirming the high level of similarity between the two isolates of *L. martiniquensis*.

Genomic variation analysis revealed 168,538 SNPs and 26,440 indels in the genome of the southern isolate PCM2 of *L. orientalis* compared to the northern isolate LSCM4, and 5736 SNPs and 4637 indels in the genome of the southern isolate PCM3 compared to the northern isolate LSCM1 of *L. martiniquensis*. The majority of the identified SNPs (422,800 SNPs, 89.74%) and indels (68,784 indels, 89.42) of the southern isolate PCM2 appeared in the upstream, downstream, intergenic, and UTR regions. In comparison, 1671 SNPs (0.36%) and 6954 indels (9.04%) could affect splice sites, start, and stop codons. Thirty-two thousand eight hundred and nine identified SNPs had a missense effect, which caused the substitution of different amino acids in the resulting proteins. For the southern isolate PCM3 of *L. martiniquensis*, most of the identified SNPs (9653 SNPs, 92.66%) and indels (13,711 indels, 88.60%) appeared in the upstream, downstream, intergenic, and UTR regions, while 44 SNPs (0.42%) and 1621 indels (10.47%) could affect splice sites, start, and stop codons, and 490 SNPs had a missense effect. The high number of genomic variations in the genome of the southern isolate of *L. orientalis* supported the observation in Figure 1 and highlighted the greater genetic distance between the two *L. orientalis* isolates compared to those of *L. martiniquensis*.

### 3.2. Comparative Functional Genomics of the Southern Isolates of L. orientalis and L. martiniquensis to Those of Other Leishmania

To gain more functional information, predicted proteomes of *L. orientalis* isolate PCM2 and *L. martiniquensis* isolate PCM3 were compared with the proteomes of the northern isolates LSCM1 and LSCM4, and those of 14 other *Leishmania* species, including *L. major*, *L. donovani*, *L. mexicana*, *L. infantum*, *L. aethiopica*, *L. brazilliensis*, *L. panamensis*, *L. chagasi*, *L. amazonensis*, *L. guyanensis*, *L. arabica*, *L. enriettii*, *L. tropica*, and *L. lainsoni*. Comparative results revealed different levels of protein sequence identity between isolates and species (Figure 2). Patterns of the protein sequence variations allowed the proteins to be clustered into six groups. The second group had the largest number of protein members (5371 proteins), followed by the third (953 proteins), first (692 proteins), fifth (520 proteins), fourth (245 proteins), and sixth (47 proteins) groups, respectively. These patterns separated *Leishmania* species into two major clusters, including a small cluster of the two isolates of *L. orientalis* and *L. enriettii*, and a large cluster of the remaining species. The large cluster was further divided into four sub-clusters, and one of these sub-clusters was the group of the two *L. martiniquensis* isolates, which were closely similar to the sub-cluster of *L. mexicana*, *L. major*, *L. infantum*, *L. donovani*, *L. panamaensis*, and *L. braziliensis*. The protein sequence difference between the two *L. orientalis* isolates was shown in groups 1, 2, 3, and 4, while the protein variation between the two *L. martiniquensis* isolates was visible in all six groups (Figure 2). These observations indicated within-species variations between isolates of the two *Leishmania* species in Thailand.

Functionally annotated proteins from the southern isolates of *L. orientalis* (PCM2) and *L. martiniquensis* (PCM3) were compared with the northern isolates (LSCM1 and LSCM4) and other Leishmanial species. Four categories (biological process, cellular component, molecular function, and enzymatic function) of the functional GO terms were assigned to proteins in each group, and the percentage of the annotated proteins is displayed in Table 2. More than 50% of the proteins in groups 2 and 5 were functionally assigned (66.15% and 51.35%), while less than 50% were assigned in groups 1, 3, 4, and 6. The overall percentage of the protein functional annotation was at 56.23%, which indicated that several *Leishmania* proteins remain hypothetical and uncharacterized (Table 2).

Proteins in the first group mostly had functions in cell adhesion (17.02%), transmembrane transport (14.89%), and protein phosphorylation (9.57%). A high percentage of proteins with metalloendopeptidase (14.04) and ABC-type transporter (11.40) activities were found in this group, while most of the first group proteins were located in the plasma membrane (74.82%) and cilia (7.18%) (Appendix A). Proteins in the first group clearly showed differences between the two isolates of *L. orientalis* and *L. martiniquensis* (Figure 2). Three proteins were present in *L. orientalis* isolate PCM2, but not the isolate LSCM4 (Table 3). Only one protein was annotated as receptor-type adenylate cyclase A.

Proteins in the second group had major functions in the processes of protein phosphorylation (5.43%), proteolysis (3.08%), microtubule-based movement (2.18%), and translation (2.04%) (Appendix A). A high percentage of the proteins in this group are located at the plasma membrane (19.43%), cytoplasm (11.75%), nucleus (8.40%), and mitochondrion (4.75%). This group had a higher proportion of proteins with ATP (3.87%), RNA (3.84%), and metal ion (3.70%) binding activities, protein kinase activity (3.53%), and structural constituents of ribosome (2.04), indicating their major involvement in the core metabolisms of the parasite. A high degree of protein sequence variation was observed when comparing the proteins of the two isolates of *L. orientalis* and *L. martiniquensis* (Figure 2). Sixty-seven proteins were present only in *L. martiniquensis* isolate PCM3 and not in the isolate LSCM1, and 32 of these proteins were annotated (Table 4). For the isolate PCM2 of *L. orientalis*, 47 proteins were absent from the isolate LSCM4; 34 of these proteins were functionally assigned (Table 4).

Major functions of the proteins in the third group included protein phosphorylation (8.37%), transmembrane transport (5.42%), proteolysis (3.45%), lipid metabolic process (2.96%), and cilium assembly (2.46%) (Appendix A). More than 50% of the proteins in this group (51.45%) were annotated to be an integral component of the membrane, cytoplasmic (9.30%), and nuclear (4.65%) proteins. A high proportion of the proteins in this group had protein kinase activity (6.61%), metal ion (12.32%) and RNA (3.96%) binding activities, and hydrolase activity (2.64%). A difference between the third-group proteins was also observed between the two isolates of *L. orientalis* and *L. martiniquensis* (Figure 2). Twenty-one proteins were present only in *L. martiniquensis* isolate PCM3, not in the isolate LSCM1, and six of these proteins were annotated (Table 4). Three proteins were present in *L. orientalis* isolate PCM2, but not the isolate LSCM4 (Table 4). Only one protein was annotated as a putative trypanothione synthetase.

The fourth-group proteins had a greater percentage of biological processes involved in translation (5.38%) and mRNA splicing via spliceosome (3.23%). The proteins of this group were mainly the integral components of the membrane (23.79%), nucleus (4.96%), and cytoplasm (3.55%). A higher proportion of the proteins in the fourth group had RNA (6.74%), metal ion (6.74%), and ATP (4.49%) binding activities and were structural components of the ribosome (4.49%). The difference between the fourth-group proteins was visible between the two isolates of *L. orientalis* and *L. martiniquensis* (Figure 2), but these proteins were not unique to either species (Table 3).

The fifth-group proteins were highly similar between the two isolates of *L. orientalis* and *L. martiniquensis* (Figure 2), and there were no *L. orientalis*- or *L. martiniquensis*-specific proteins (Table 3). Interestingly, the proteins in this group of *L. orientalis* were quite distinct from those of *L. enriettii*. The fifth-group proteins had a high percentage of functions related to protein phosphorylation (6.63%), translation (3.06%), and protein dephosphorylation (2.55). The proteins of this group appeared more in the membrane (25.23%), cytoplasm (10.51%), and nucleus (6.91%). These fifth-group proteins had a high proportion of metal ion binding activity (6.19%), protein kinase activity (4.76%), ATP (4.29%) and RNA (3.33%) binding activities, and ATP hydrolysis activity (3.33%).

The sixth group was the smallest group, which had five assigned biological processes with the same percentage, including 7-methylguanosine mRNA capping, cellular amino acid metabolic process, Golgi vesicle transport, microtubule-based movement, and phosphorylation. More than 70% of the proteins in this group were located at the membrane (72.73%) and cytoplasm (18.18%). A high proportion of the sixth-group proteins had metal-binding activity (25%) and hydrolase activity (25%). Two unique proteins were present in *L. martiniquensis* isolate PCM3, but not the isolate LSCM1 (Table 3), and none of them were annotated.

## 4. Discussion

This study successfully decoded the drafted genomes of *L. orientalis* isolate PCM2 and *L. martiniquensis* isolate PCM3 from the southern province in Thailand by relying on the short-read sequencing technology. The genomes of these southern isolates were compared with those of the northern isolates LSCM4 and LSCM1, recently published by Almutairi et al. [20], which employed a combination of short- and long-read sequencing platforms, providing excellent genomic references for this study. When assembled and compared to the references, our drafted genomes displayed an excellent coverage level and depth, confirming the same species relatedness. However, the genetic distance between the northern and southern isolates of these two species was revealed to be significant by integrating results from the comparative genomes, variant analysis, and proteome comparison. This genomic distinctiveness could be due to the geographical distance of the northern and southern provinces, which was at least 1500 kilometers away. The greater structural genomic and variant difference in the two *L. orientalis* isolates could be supported by the previous leishmaniasis situation in Thailand wherein *L. orientalis*, previously recognized as *L. siamensis*, had been neglected and was thought to be *L. martiniquensis* for several years until the past ten years [9].

Comparative proteome and functional annotation of the southern isolates of *L. orientalis* (PCM2) and *L. martiniquensis* (PCM3) with those of the northern isolates (LSCM1 and LSCM4) and another 14 *Leishmania* species showed a close relationship between the two isolates. The clustering of *Leishmania* species shared similar phylogenetic patterns with previous genetic studies, such as the separation of *L. lainsoni* and *L. guyanensis* from *L. donovani*, *L. major*, *L. tropica*, and *L. mexicana* based on the *hsp*70 phylogeny [34]. The protein sequence similarity-based clustering gave six protein groups and could depict an overview of the functional relatedness of all proteins across *Leishmania* strains and species (Figure 2). The more distant the relationship, the greater protein profile variation would be observed. The highly conserved proteins in part of the second group could be involved in the core processes essential to the survival of these parasitic species. The proteins with lower sequence identity, such as those in groups 1, 3, 4, and 5, could reflect the molecular adaptation of the southern isolates of *L. orientalis* (PCM2) and *L. martiniquensis* (PCM3) to maintain their survival under different environments, consistent with the study of Salloum et al. [35], which identified SNPs in proteins related to iron acquisition, sterol synthesis, and drug resistance and proteins specific to a different population of *L. tropica*. The strain-specific proteins in Table 4 contained proteins associated with virulence (amastin, other surface glycoproteins, heat shock protein [36], and pyroglutamyl-peptidase I [37]), drug resistance (putative beta-lactamase and a myriocin drug target, palmitoyltransferase [38]), and the intracellular oxidative stress response (putative trypanothione synthetase [39]), as well as several other uncharacterized proteins, which could provide additional information on these southern isolates (PCM2 and PCM3) and their pathogenicity.

The genomic variations (Figure 2) and the emergence of strain-specific proteins (Table 3 and Table 4) could drive the species diversification process, allowing the parasites to acclimatize and adapt to different climates and environments between the northern and southern parts of the country, perhaps suggesting the separation into two different strains of both *Leishmania* species. This statement could be supported by the metaphylome study of *Leishmania* proteins, which found the expansion of an important surface glycoprotein family, named amastin [40], which was additionally found in the southern isolates (PCM2 and PCM3) of *L. orientalis* and *L. martiniquensis* in Thailand (Table 4). Therefore, the emergence and separation of these northern (LSCM1 and LSCM4) and southern (PCM2 and PCM3) strains could give alarming calls for more genomic and population investigation and close monitoring of *L. orientalis* and *L. martiniquensis* in Thailand and surrounding Southeast Asian regions.

## 5. Conclusions

In conclusion, this study successfully sequenced the draft genomes of the southern isolates (PCM2 and PCM3) of *L. orientalis* and *L. martiniquensis* in Thailand, and the comparison with the genomes of the northern isolates (LSCM1 and LSCM4) revealed species-level similarity with a level of genome and proteome variation, suggesting the different strains. Comparing the proteins of these southern strains to those of the northern ones and 14 other *Leishmania* species showed six protein groups with a number of unique proteins: 53 for the southern strain PCM2 of *L. orientalis* and 97 for the strain PCM3 of *L. martiniquensis*. These proteins were related to virulence, drug resistance, drug target, and the stress response, which could be targeted for further experimental characterization. Therefore, the findings could indicate the need for more genetic and population genomic investigation and close monitoring of *L. orientalis* and *L. martiniquensis* in Thailand and neighboring regions.

## Figures and Tables

**Figure 1 biology-11-00515-f001:**
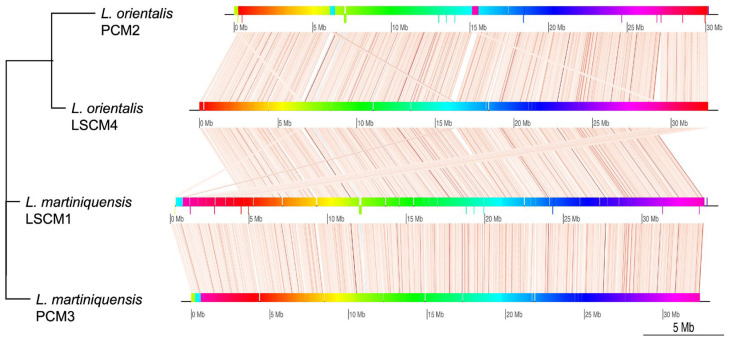
Alignment of genomic contigs from *L. orientalis* southern isolate PCM2 and *L. martiniquensis* southern isolate PCM3 compared to those of the northern isolates LSCM1 and LSCM4 in Thailand. Colored shades represented the conserved synteny, and the bar height indicates the level of sequence identity. Lines show associated conserved syntenic blocks across the compared isolates.

**Figure 2 biology-11-00515-f002:**
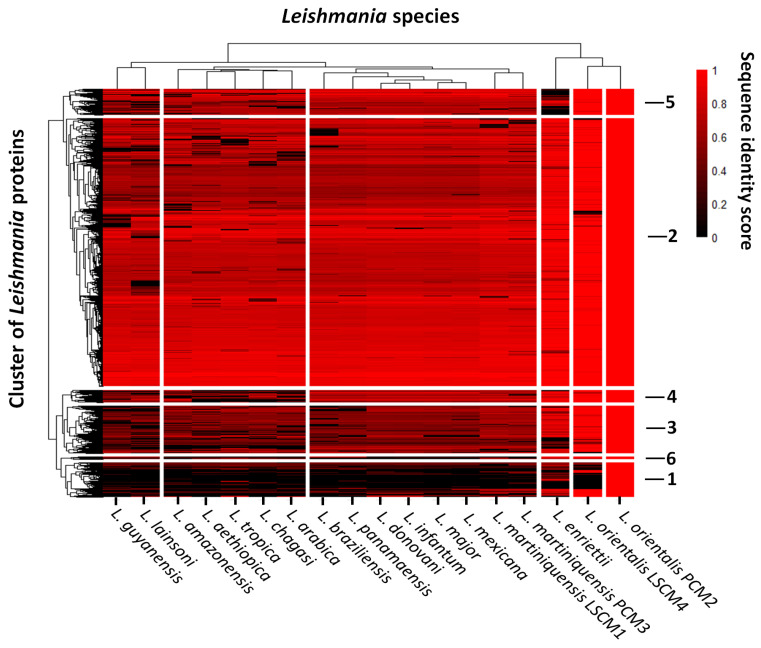
Comparison of protein sequences of the southern isolates of *L. orientalis* PCM2 and *L. martiniquensis* PCM3 in Thailand with the northern isolates and other 14 *Leishmania* species. The protein identity scores were hierarchically clustered into two clusters of *Leishmania* species (depicted by the column dendrogram) and six groups of proteins (represented by the row dendrogram). Color scale represents the level of sequence identity scores. Numbers labeled on the right of the heatmap indicate the group numbers (1 to 6).

**Table 1 biology-11-00515-t001:** Basic characteristics of draft genomes of *L. orientalis* isolate PCM2 and *L. martiniquensis* isolate PCM3 from the southern part of Thailand.

Genome Characteristics	*L. orientalis* PCM2	*L. martiniquensis* PCM3
Contig number	4399	42
Total length	30.01 Mbp	32.39 Mbp
N50	14.03 Kbp	1.05 Mbp
L50	651	11
Number of coding genes	8989	9577
%GC	59.02	59.92

**Table 2 biology-11-00515-t002:** Comparison of the number of uniquely assigned GO terms (biological process, cellular component, molecular function, and enzymatic function) to Leishmanial proteins and percentage of protein annotation in the six groups clustered according to the protein sequence similarity between the southern isolates of *L. orientalis* (PCM2) and *L. martiniquensis* (PCM3) and the northern ones (LSCM1 and LSCM4), as well as other *Leishmania* species.

Protein Groups	Number of Uniquely Assigned GO Terms (Percentage of Annotated Proteins)				Total Number of Proteins in Each Group (Percentage of Overall Annotated Proteins)
	Biological Process	Cellular Component	Molecular Function	Enzymatic Function	
1	43 (13.58)	23 (20.09)	59 (16.47)	30 (8.96)	692 (21.24%)
2	762 (53.81)	378 (72.07)	629 (54.85)	474 (26.57)	5371 (66.15%)
3	119 (21.30)	71 (36.10)	125 (23.82)	61 (10.07)	953 (33.26%)
4	77 (37.96)	64 (57.55)	65 (36.33)	24 (11.84)	245 (44.49%)
5	136 (37.69)	88 (64.04)	109 (40.38)	58 (14.81)	520 (51.35%)
6	5 (10.64)	3 (23.40)	6 (17.02)	3 (8.51)	47 (19.15%)
Total					7828 (56.23%)

**Table 3 biology-11-00515-t003:** Comparison of the annotated proteins that were found only in *L. orientalis* and *L. martiniquensis*. The compared numbers are depicted as the proteins uniquely identified in either of the two isolates or both. The “-” sign indicates the absence of the unique proteins in that isolate.

Protein Cluster	Number of Unique Proteins Only Found in Either *L. orientalis* or *L. martiniquensis*					
	*L. orientalis*			*L. martiniquensis*		
	Unique to PCM2	Unique to LSCM4	Found in Both Isolates	Unique to PCM3	Unique to LSCM1	Found in Both Isolates
1	3	-	28	7	-	-
2	47	-	-	67	-	-
3	3	-	-	21	-	-
4	-	-	-	-	-	-
5	-	-	-	-	-	-
6	-	-	-	2	-	-
Total	53	-	28	97	-	-

**Table 4 biology-11-00515-t004:** Unique proteins present in the proteomes of *L. orientalis* isolate PCM2 and *L. martiniquensis* isolate PCM3. The “+” in the cluster column represents the presence of the protein in *L. orientalis* isolate PCM2.

No	Gene ID	Cluster	Protein Name
1	k59_7101.g4222	1	Amastin surface glycofamily protein
2	k59_5884.g7963	1	D-amastin
3	k59_5466.g3144	1	Putative malic enzyme
4	k59_1669.g769	1+	Receptor-type adenylate cyclase a
5	k59_10223.g2223	2+	Aminoacyl-tRNA editing domain containing protein, putative
6	k59_6373.g6990	2	Aminoacyl-tRNA editing domain containing protein, putative
7	k59_6105.g6831	2	Ankyrin repeat family protein
8	k59_2596.g5206	2+	Antiviral helicase
9	k59_4730.g7538	2	ATP-dependent DNA helicase
10	k59_1985.g3871	2+	Beta-lactamase, putative
11	k59_7875.g3627	2+	C3H1-type domain-containing protein
12	k59_10386.g8210	2	Cactus-binding C-terminus of cactin protein, putative
13	k59_3265.g2893	2	Centromere/microtubule binding protein cbf5, putative
14	k59_10658.g8866	2+	Chromatin assembly factor 1 subunit A, putative
15	k59_10459.g1059	2+	CS domain-containing protein
16	k59_3314.g4482	2	Cyclin dependent kinase-binding protein, putative
17	k59_5445.g2737	2	D-3-phosphoglycerate dehydrogenase-like protein
18	k59_4821.g8945	2	Endoplasmic Reticulum-Golgi Intermediate Compartment (ERGIC)
19	k59_2500.g2688	2	Enkurin domain-containing protein
20	k59_10260.g4158	2+	Essential nuclear protein 1
21	k59_8350.g1377	2+	Exosome-associated protein 1
22	k59_9227.g4203	2	Farnesyltransferase alpha subunit
23	k59_6707.g85	2+	GB1/RHD3-type G domain-containing protein
24	k59_6854.g4013	2+	Glucose-6-phosphate isomerase (Fragment)
25	k59_199.g4824	2+	Glycosyltransferase (GlcNAc), putative
26	k59_10459.g1058	2	GOLD domain-containing protein
27	k59_4463.g6091	2+	Guanine nucleotide-binding protein subunit beta-like protein
28	k59_1181.g5237	2+	H(+)-exporting diphosphatase
29	k59_9085.g8550	2	Heat shock protein DnaJ
30	k59_457.g6730	2+	HECT domain-containing protein
31	k59_1395.g2084	2	Intraflagellar transport protein 22
32	k59_8474.g5391	2	Intraflagellar transport protein D4
33	k59_9119.g816	2	MIZ/SP-RING zinc finger family protein
34	k59_7389.g5456	2	mRNA processing protein, putative
35	k59_5206.g5478	2	MRP-L46 domain-containing protein
36	k59_8917.g2151	2	MYND zinc finger (ZnF) domain-like protein
37	k59_10044.g5171	2+	NLE domain-containing protein
38	k59_7676.g6287	2	NTF2 domain-containing protein
39	k59_5657.g997	2+	Nuclease-related domain containing protein, putative
40	k59_5439.g2470	2+	PAB-dependent poly(A)-specific ribonuclease subunit 3
41	k59_660.g3683	2	Palmitoyltransferase
42	k59_2880.g6470	2	PHD domain-containing protein
43	k59_5927.g1098	2	Phosphorylated CTD interacting factor 1 WW domain containing protein
44	k59_5927.g1098	2	Phosphorylated CTD interacting factor 1 WW domain containing protein
45	k59_10222.g2044	2+	Poly(ADP-ribose) polymerase and DNA-Ligase Zn-finger region
46	k59_9477.g3999	2+	PRK domain-containing protein
47	k59_8589.g1275	2	Protein disulfide isomerase, putative
48	k59_5413.g1718	2	Putative DNA repair protein RAD2
49	k59_9374.g604	2+	Putative flagellar radial spoke protein-like
50	k59_10153.g8013	2	Putative Gamma-soluble NSF attachment protein (SNAP-gamma)
51	k59_2626.g6421	2	Putative mitochondrial RNA binding complex 1 subunit
52	k59_7006.g1237	2+	Putative prefoldin subunit 2
53	k59_2660.g7703	2+	Putative pre-mRNA branch site protein p14
54	k59_10051.g5820	2	Putative tRNA pseudouridine synthase A
55	k59_2665.g8021	2+	Putative unspecified product
56	k59_2453.g835	2+	Putative variant surface glycoprotein
57	k59_1240.g7057	2+	Pyroglutamyl-peptidase I (PGP), putative
58	k59_10051.g5821	2	Queuosine salvage protein
59	k59_1202.g6035	2	Rab3 GTPase-activating protein catalytic subunit
60	k59_7391.g5855	2	Rab-GTPase-TBC domain containing protein, putative
61	k59_3567.g2504	2	Related to elongation factor-2 kinase efk-1b isoform-like protein
62	k59_1109.g2256	2+	RNA recognition motif family protein
63	k59_4042.g1153	2+	Roadblock/LC7 domain containing protein, putative
64	k59_3916.g6656	2+	Secretory carrier membrane protein 3
65	k59_3762.g682	2	SNARE associated Golgi protein, putative
66	k59_1957.g3007	2+	Sperm-tail PG-rich repeat family protein
67	k59_5678.g1713	2+	Succinate dehydrogenase assembly factor 2, mitochondrial
68	k59_5098.g435	2+	TFIIS central domain-containing protein
69	k59_8533.g7240	2+	Translation initiation factor
70	k59_3506.g770	2+	U5 snRNP-specific 40 kDa protein, putative
71	k59_6432.g50	2+	WD domain, G-beta repeat family protein
72	k59_6901.g5645	3	Alpha-1,3/1,6-mannosyltransferase ALG2
73	k59_4782.g8504	3	ATP-dependent DNA helicase
74	k59_6737.g659	3	ATP-dependent helicase, putative
75	k59_7325.g3419	3	N-terminal region of Chorein, a TM vesicle-mediated sorter family protein
76	k59_7204.g7945	3+	Putative trypanothione synthetase
77	k59_5425.g1966	3	RNA binding protein-like protein
78	k59_2661.g7832	3	SAT domain-containing protein

## Data Availability

Data are available on the NCBI genome database via the BioProject accession number PRJNA741905 for *L. orientalis* isolate PCM2 and PRJNA728409 for *L. martiniquensis* isolate PCM3.

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
