# Peer review of "Comparative Draft Genomes of Leishmania orientalis Isolate PCM2 (Formerly Named Leishmania siamensis) and Leishmania martiniquensis Isolate PCM3 from the Southern Province of Thailand"

_biology, 2022, doi:10.3390/biology11040515_

Round 1

Reviewer 1 Report

I carefully reviewed the manuscript titled "Comparative draft genomes of Leishmania orientalis isolate PCM2 (formerly named L. siamensis) and Leishmania martiniquensis isolate PCM3 from the southern province of Thailand". 

The study here reported is interesting and it gave important information about the situation of Leishmaniasis in Thailandia which should be addressed in the nearly future. 

Despite it, the manuscript before being considered for its publication needs major changes. Particularly the introduction needs to be improved. 

Here some example

Page 2, line 75: “Autochthonous VL cases have continuously been reported since 1996 in the Southern province [10] and northern province in 2012 [11].” The authors are referring to Thailand but the sentence is not clear. I should be edited.

Page 2, Lines 76-100 “Leelayoova et al. [8,12] … … … diagnostic tests and drugs” this paragraph needs an important edit. Authors report mostly the same information repeatedly and do not emphasize the key information.  

Pag 3 , line 108 “ … and L. peruviana.” A reference is missed.

Here some minor changes

AFFILIATION:

The affiliation for each author is not correct. Please check it and correct it.

METHODS:

Pag4, line 190 “…similar to the values reported from other Leishmania species (Table 1).” A reference is missed here

RESULTS: 

Page 4, line 191 “ … similar to those of the northern isolates of L. orientalis 192 isolate LSCM4 (8,162 genes), and L. martiniquensis isolate LSCM1 (7,993).” A reference is missed here.

Page 4, lines 155-156 terms “southern isolates” and “northern samples” are confusing. I would suggest to authors identify their isolated from the references naming them differently.

Reviewer 2 Report

The manuscript “Comparative draft genomes of Leishmania orientalis isolate PCM2 (formerly named L. siamensis) and Leishmania martiniquensis isolate PCM3 from the southern province of Thailand” by Anuntasomboon and colleagues shows genome sequencing of two important Leishmania species in the Southern part of Thailand. The results are well detailed and written. The results are well detailed and written. The authors found several unique proteins in the isolates (53 for L. orientalis (PCM2) and 97 for L. martiniquensis (PCM3)) and these proteins are related to virulence, drug resistance, intracellular oxidative stress response and other uncharacterized proteins. Studies like this are important because provide a better understanding about species diversity/ pathogeneses of parasites related to human health.

I have a few comments below:

Line 22: Please, the first time the species name is mentioned, write the full name: “Leishmania orientalis” and “Leishmania martiniquensis”.

Line 34: “L. orientalis (formerly named Leishmania siamensis)”

Line 56: Macrophages are the main host cells of Leishmania and are the main cells in the initial response to infection, however, they are not the only cells parasitized. Please make this clear in the text.

Line 62: Please complete your sentence - Manifestations can be grouped into cutaneous/tegumentary leishmaniasis (CL) and visceral leishmaniasis (VL) form. CL can be further classified into several manifestations: Localized cutaneous leishmaniasis (LCL), mucosal leishmaniasis (ML), disseminated leishmaniasis (DL), and diffuse cutaneous leishmaniasis (DCL).

Line 63: ”leishmaniases” - the correct is leishmaniasis

Line 63/64: Please describe more about CL types.

Line 75: “…since 1996 in the Southern province [10] and northern province in 2012 [11]” – Please exclude the first “province” and write in lowercase “southern”

Reference 10 is incomplete. The name of the Journal, pages and year are missing.

Line 76: Please exclude “Leelayoova et al. [8,12] reported that” and put just the reference number at the end of the sentence

Line 77: As we are in the introduction, you can write the full name of the species (the first time that are mentioned) - “L. enriettii” please write Leishmania enriettii

Line 78: “L. siamensis” and “L. orientalis” - please write “Leishmania siamensis” and “Leishmania orientalis”

Line 79: “L. martiniquensis” – please write “Leishmania martiniquensis

Line 81: “L. martiniquensis and L. orientalis can be the cause of both CL and VL” - What type of cutaneous manifestation do these protozoans cause?

Line 82: Is “Trang” the name of the lineage? If so, please write “Trang” in line 78 (lineage TR, isolate PCM2).

Line 82/83: Please uppercase “s” from “AIDs patience”

Line 84/85: Leishmania donovani, Leishmania lainsoni, and Leishmania major

Line 85: Previous reviews by our group [8] showed that …

Line 94: Please remove “reviewed by Leelayoova et al. [8]” and reference at the end of the sentence

Line 97: I particularly do not approve of this type of direct citation of the authors in the text. If possible, please rephrase the sentence without citing "...by Mungthin et al. ..."

Line 103/104: “L. infantum” and “L. braziliensis”– please write “Leishmania infantum” and “Leishmania braziliensis

Line 104: Leishmania in italic

Line 104: Symptomatic infections by L. braziliensis usually lead to LCL, and sometimes to ML.

Line 112: Leishmania in italic

In the methodology, the authors mention that PCM2 and PCM3 isolates were obtained from patients. Was this collection approved by any Human Research Ethics Committee? If so, please cite in the text.

Before starting the genome sequencing studies, did you test by PCR (or other technique) the different isolates that you had? How did you know they were different species? Please include this information in the text.

Line 192: Please replace “similar” by “while”  

Line 220: In Figure 1, please fix the spelling of “L. oreintalis”

Line 178/247/Figure 2: please fix the spelling of “L. amezonensis”. The correct is amazonensis

Line 257/Figure 2: please fix the spelling of “L. maxicana”. The correct is mexicana
